# Assessing Coastal Flood Susceptibility in East Java, Indonesia: Comparison of Statistical Bivariate and Machine Learning Techniques

Entin Hidayah [1,*], Indarto [2], Wei-Koon Lee [3], Gusfan Halik [1] and Biswajeet Pradhan [4,5,6]

1   Hydrotechnical Laboratory, Department of Civil Engineering, University of Jember, Jalan Kalimantan No 37, Jember 68121, Jawa Timur, Indonesia
2   Department of Agricultural Engineering, University of Jember, Jalan Kalimantan No 37, Jember 68121, Jawa Timur, Indonesia
3   School of Civil Engineering, College of Engineering, Universiti Teknologi MARA, Shah Alam 40450, Selangor, Malaysia
4   Centre for Advanced Modelling and Geospatial Information Systems (CAMGIS), School of Civil and Environmental Engineering, University of Technology Sydney, Ultimo, NSW 2007, Australia
5   Center of Excellence for Climate Change Research, King Abdulaziz University, P.O. Box 80234, Jeddah 21589, Saudi Arabia
6   Earth Observation Centre, Institute of Climate Change, Universiti Kebangsaan Malaysia (UKM), Bangi 43600, Selangor, Malaysia
*   Correspondence: entin.teknik@unej.ac.id; Tel.: +62-813-3030-0604

**Abstract:** Floods in coastal areas occur yearly in Indonesia, resulting in socio-economic losses. The availability of flood susceptibility maps is essential for flood mitigation. This study aimed to explore four different types of models, namely, frequency ratio (FR), weight of evidence (WofE), random forest (RF), and multi-layer perceptron (MLP), for coastal flood susceptibility assessment in Pasuruan and Probolinggo in the East Java region. Factors were selected based on multi-collinearity and the information gain ratio to build flood susceptibility maps in small watersheds. The comprehensive exploration result showed that seven of the eleven factors, namely, elevation, geology, soil type, land use, rainfall, RD, and TWI, influenced the coastal flood susceptibility. The MLP outperformed the other three models, with an accuracy of 0.977. Assessing flood susceptibility with those four methods can guide flood mitigation management.

**Keywords:** coastal flood mapping; frequency ratio; weight of evidence; random forest; multi-layer perceptron

## 1. Introduction

Climate change affects the occurrence and increase in the frequency, intensity, spatial area, duration, and time of extreme rain events that impact the occurrence of extreme floods [1]. These flood events can cause severe damage to life, habitat, infrastructure, and property and directly impact the economy and social sectors [2,3]. Flood susceptibility assessments seek to understand the key factors that cause flooding and the ability to determine the location, likelihood, and severity of flooding to facilitate flood mitigation measures.

Ideally, hydrological and hydraulics modeling can produce a detailed flood hazard map in a flood hazard assessment [4]. Nevertheless, non-linear hydrological modeling methods are challenging to implement because of the complexity of the catchment area [5], and thus, are often limited to small-scale applications [6,7]. Furthermore, limited data availability, especially in developing countries, makes it challenging to accurately model local-scale flood susceptibility [8].

Developing flood susceptibility assessment techniques using a geographical information system (GIS) related to geomorphological factors is an alternative solution. Various

factors, including morphological settings, land use, rainfall intensity, lithology, and flood events, are currently being developed in flood susceptibility modeling [9]. These factors are known to cause flooding, but individual factors have a different degree of influence depending on the characteristics of each location. Based on [10], the coastal long-period wave is also related to coastal water-body inundation or flooding. It was shown that the most influential factors are elevation [11], slope [8], land use [12], and the normalized difference vegetation index (NDVI) [13].

Various probabilistic models have been applied to map flood susceptibilities, such as bivariate statistics, including frequency ratio (FR) [8], the logistic regression method [14], the weight of evidence (WofE) statistical index, and Shannon's entropy [15]. The RF method was reported to produce good accuracy [8]. The WofE method was reported to have similar accuracy to the FR method [16]. These two methods look for correlations between flood events and the flood-triggering factor, and thus, are very efficient for flood prediction [17]. The selection of the correct model weight can significantly affect the prediction accuracy. However, several studies showed that apart from the equal weightage method, differences in the factor conditions may affect the accuracy of flood prediction [18].

In the latest developments, flood susceptibility modeling using machine learning produced outstanding accuracy; this modeling consisted of random forest (RF) [19], logistic regression (LR), support vector machine (SVM), and multi-layer perceptron (MLP) methods [9]. In addition to the abovementioned machine learning methods, the RF [19], SVM [18], and MLP [9] also provide more accurate results than other's machine learning models for mapping flood susceptibility. MLP has high stability while having a smaller structure than most other neural networks [20]. RF is a feasible way to overcome the problem with multiple matching of regression trees that it tends to outfit the training dataset, and thus, performs inadequately when given an uncertain dataset [18]. Based on [21], the RF model has a strong goodness of fit, where the forecasted outputs are relatively close to the actual outputs.

According to the literature mentioned above, the FR and WofE models from the bivariate statistics method have linear correlations between flood events and the flood-triggering factor, and thus, are very efficient for flood prediction [16]. Furthermore, in machine learning methods, MLP models are highly capable of modeling the non-linear relationship between an explanatory variable and the target variable of flood susceptibility [9]. RF is an ensemble classifier system that is based on binary decision trees [22]. It can easily handle a large number of variables and it is a statistically-based approach. So far, machine learning in susceptibility mapping has only focused on certain aspects that cannot portray the correlation between one factor and another. Therefore, it is necessary to conduct comprehensive research to determine what factors influence the coastal flood susceptibility level.

This study aimed to map flood susceptibility and analyze the influencing factors in coastal areas regionally in the Pasuruan and Probolinggo Regencies. A selection of FR and WofE models from bivariate statistical methods and RF and MLP from machine learning methods were taken as good models to implement flood vulnerability mapping to explore the results of flood susceptibility mapping in order to comprehensively mitigate flooding. The factors were evaluated using a multi-collinearity test and information gain ratio. One of the other objectives of this study was to determine which factors had the highest impact on and conditioning factor from each model for flood susceptibility in the coastal area. Known flood conditioning factors can be used to minimize the impact of efficient factors. Therefore, this study examined each of these methods by using spatial data at a regional scale in order to produce coastal flood susceptibility maps.

## 2. Materials and Methods

### 2.1. Study Area

The study area was located at the Gembong–Pekalen basin in the Probolinggo and Pasuruan Regencies, covering 824.921 km$^2$. Geographically, it was located at 8°00′ S to 7°30′ S and 112°45′ E to 113°30′ E, as shown in Figure 1. The basin comprises many small

watersheds that flow into the Java Sea. The basin is formed by three geomorphological units, namely, Mount Arjuno, Bromo, and Argopuro. The highest elevation in this area is 3323 m above sea level (m.a.s.l). The upstream part of the watershed has a steep slope, while the downstream part is relatively flat.

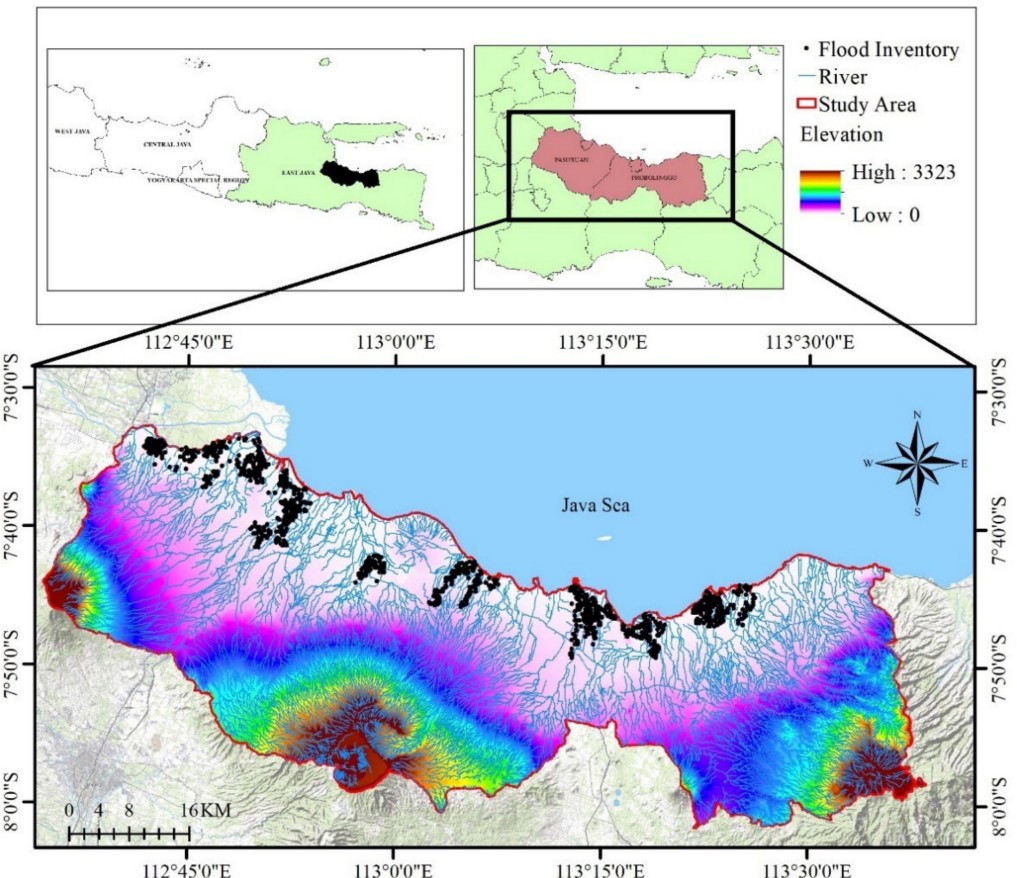

**Figure 1.** Study area map.

Based on the Water Resources and Public Works Department, East Java, Indonesia, this region has a tropical climate characterized by high rainfall, temperature, and humidity. This region experiences two seasons: the rainy season, which usually occurs from December to March, and the dry season from May to October. The maximum daily rainfall for the period 2000–2019 was 243 mm. Long-term average temperatures varied from 21 °C to 33 °C by region, except for the mountainous areas at high elevation, where temperatures are usually cooler.

*2.2. Methodology*

The methodology used in this flood susceptibility assessment consisted of four stages, as shown in the flowchart (Figure 2). The first was to compile the inventory and map the history of flood locations. The second was selecting coastal flood conditioning factors using a multi-collinearity test and information gain ratio. The third was constructing a flood susceptibility model by applying bivariate statistics and machine learning based on a data training technique. The fourth was to measure the performance of the flood susceptibility model based on the area under curve (AUC) value. Finally, the performance measurement of the flood susceptibility model was based on the testing data sets using the area under curve (AUC) value. Meanwhile, the RF and MPL process was determined using the Waikato Environment for Knowledge Analysis (WEKA) program.

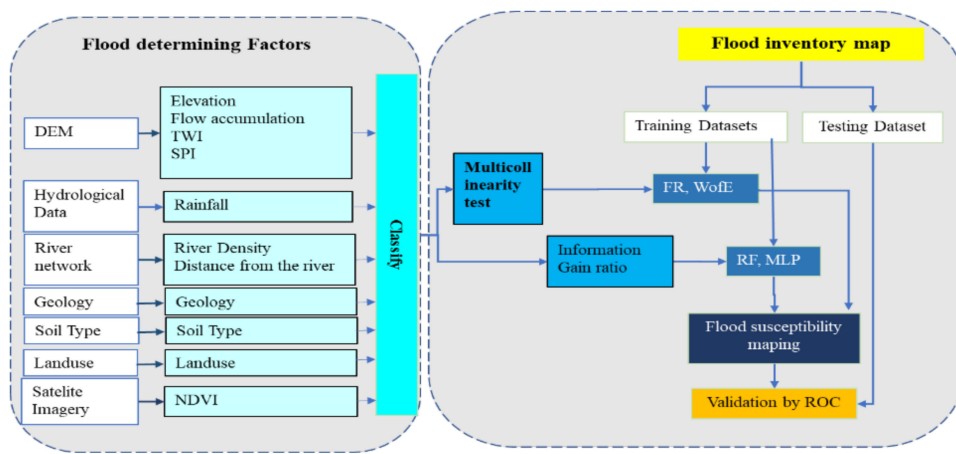

**Figure 2.** Flow chart of the research.

### 2.2.1. Flood Inventory

In this study, flood data were inventoried from the results of digitizing periodic flood polygons from 2010 to 2020, which were collected from the Technical Implementation Unit of Water Resources Management, Pasuruan Regency. Next, ground checkpoints were carried out to verify the history of the flood incident. The results of the ground check data of 2368 points (Figure 1) were divided into two sets: 70% for training and 30% for validation [9,15].

### 2.2.2. Flood Conditioning Factors

This study selected eleven flood conditioning factors that are commonly used in flood modeling [22,23]. The factors used are numerical and categorical, consisting of seven layers of digital elevation model (DEM) to generate elevation, flow accumulation, TWI, and SPI; land use; river network generating the river density and distance to the river; soil; geology, satellite imaging generating the normalized difference vegetation index (NDVI); and rainfall derived from hydrometeorology. The 11 factors consisted of interval factors classified into the following: altitude, flow accumulation (FA), topographic wetness index (TWI), stream power index (SPI), NDVI, distance to the river, river density, rainfall, land use, soil, and geology. In detail, the reduction of factor, source, and resolution data or scale can be seen in Table 1. Furthermore, the data processing was converted into a raster form to equalize the resolution. The data used are from adjacent time ranges.

**Table 1.** Flood conditioning factors data.

| Layer | Factor | Source | Resolution/Scale |
|---|---|---|---|
| DEM | Elevation | USGS Explore | 30 × 30 m |
| | Flow accumulation | | |
| | TWI | | |
| | SPI | | |
| Landsat 8 imagery | NDVI | USGS, 2020 | 30 × 30 m |
| River network | River density | Rupa Bumi Indonesia | 1:25,000 |
| | Distance to the river | | |
| Hydro-meteorology | Rainfall | East Java Provincial Public Works Service | 1:25,000 |
| Soil | Soil | ESDM Department | 1:250,000 |
| Geology | Geology | ESDM Department | 1:250,000 |
| Land use | Land use | Rupa Bumi Indonesia | 1:25,000 |

The elevation is the most pertinent flood conditioning factor [23]. Most flood events are located in a low-elevation area. Hence, elevation is chosen as an essential factor that must be included in the model. Each class has the same proportion, ranging from 16.58% to 16.90%. The lower the elevation class, the slightly larger the proportion. Flow accumulation is the flow concentration, which is one factor that affects flood susceptibility [24]. It is derived from the DEM, which is the sum of the pixel streams of the surrounding pixels indicating the runoff zone. TWI is a factor that is widely used to indicate the tendency of accumulation of water flow at a certain point in the catchment area influenced by the slope gradient. The TWI value is very highly correlated with the flood conditioning factor [17]. The TWI value is calculated using Equation (1):

$$TWI = \ln\left[\frac{A_s}{\tan\beta}\right] \tag{1}$$

where $A_s$ is the upstream contribution area and $\beta$ is the slope angle value. In Figure 3, the TWI classification has almost the same proportion value for all classes, namely, the range between 15.52% to 18.81%. SPI is related to the strength of the flow in the watershed. The value of the SPI index can be calculated using Equation (2):

$$SPI = \alpha\tan\beta \tag{2}$$

where $\alpha$ represents the total slope area flowing through a point (m$^2$/m) and $\beta$ represents the slope angle value. SPI classification in the first two classes was the dominance of the SPI values, from the lowest, namely, 40.81% and 57.53%.

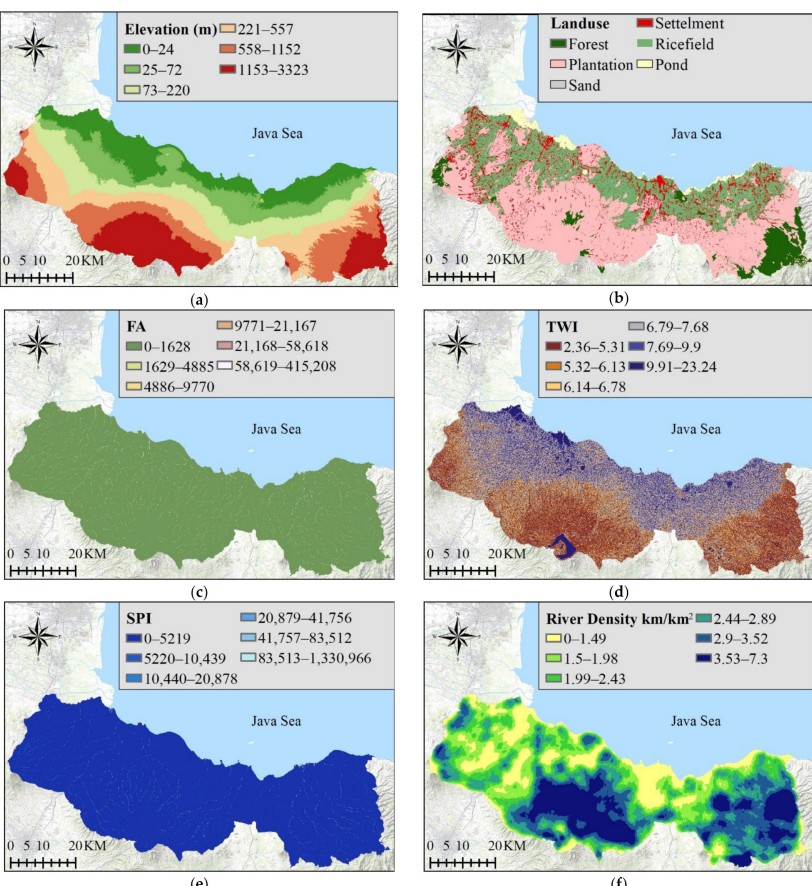

**Figure 3.** *Cont.*

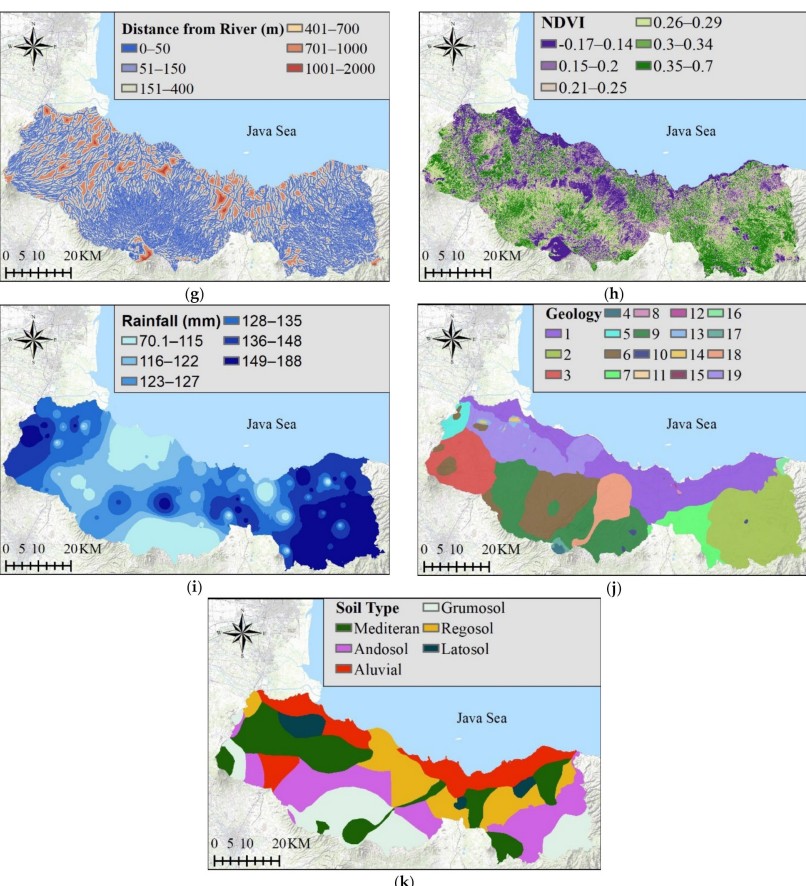

**Figure 3.** Input thematic layers: (**a**) elevation, (**b**) land use, (**c**) flow accumulation (FA), (**d**) topographic wetness index (TWI), (**e**) stream power index (SPI), (**f**) NDVI, (**g**) distance to rivers (DR), (**h**) river density (RD), (**i**) rainfall intensity, (**j**) soil type, and (**k**) geology.

To accurately calculate the amount of vegetation (canopy), the normalized difference vegetation index (NDVI) was used and can be calculated [19] using Equation (3):

$$NDVI = \frac{(NIR - RED)}{(NIR + RED)} \qquad (3)$$

where NIR is the reflectance value of the infrared channel and RED is the reflectance value of the red channel. The NDVI classification for the lowest value was negative (−0.167) and the highest value was 0.705. NDVI data collection was carried out on 13 June 2020.

The network density is the result of a calculation that involves dividing the flow length (km) by the area of the watershed (km$^2$). In Figure 3, the river network density in the study area for each class had almost the same value, i.e., between 16.12 km/km$^2$ and 19.17 km/km$^2$. The distance to a river can determine the level of susceptibility of an area to flooding. The closer the area is to a river, the more prone to flooding.

Rainfall is an important factor in triggering flood susceptibility because when there is high-intensity rainfall, the potential for flooding is even greater. The rainfall intensity used here is a 5-year return period according to the conditions of flood events in the study area. The determination of the return period of rainfall using the appropriate distribution in Indonesia, namely, normal, Gumbel, log-normal, or log-Pearson, which was then calculated using frequency analysis. The selection of the distribution according to the characteristics of the data using the chi-square test. After obtaining the design rainfall height, the regional average rainfall was determined using the inverse distance weighted method [25]. The rainfall classification ranged from 70.1 to 188 mm. The highest grades were generally in mountainous areas, and the lowest rates were in coastal areas.

Soil type is one of the parameters used to determine flood modeling parameters. Based on the Water Resources and Public Works Department, East Java, Indonesia, soil types at the research site consisted of 6 categories, with the largest total area being Regosol, followed by Mediterranean, Andosol, Grumusol, Latosol, and Alluvial. In Figure 3, the geological classification consists of 19 categories. Six types of geology dominate in the study area, namely, alluvial, Argopuro volcanic rock, Pandak volcanic rock, Lamongan volcanic rock, and Rabanno tuff, with proportion values of 20.87%, 18.98%, 15.63%, 11.82%, and 11.63% respectively. Land use is one of the critical factors when calculating flood modeling that affects the runoff coefficient. In Figure 3, land-use data was compiled based on the Indonesian Earth Map with a scale of 1:25,000, which was revised in 2013 and updated with the Landsat 8 satellite image base map on 13 June 2020. The land-use data were grouped into six classes, namely, forests, gardens, sand, settlements, rice fields, and rivers.

Multi-collinearity will significantly affect the results if there are multiple independent factors. Multi-collinearity occurs when there is a high correlation between the conditions [26]. In this study, a multi-collinearity test was used to select the flood conditioning factors. The Pearson correlation test is part of the multi-collinearity test among the independent causal factors. The Pearson correlation coefficient value of 0.70 was considered the threshold for multi-collinearity [27].

The feature selection process was used to select the factors that were used in the MLP and RF. Flood events are influenced by many complex and generally interrelated factors. Therefore, an integrated analysis of the relationship between the factors that affect flooding needs to be done. In addition, attribute selection is an important phase in data mining preprocessing according to [28]. To select a subset of the actual attribute space based on the ability of the discriminant to improve data quality, the feature selection method used the information gain ratio. The formula for the information gain ratio is shown in Equation (4):

$$\text{Entropy } (S) = -\sum_{i=1}^{n} p_i * \log_2(p_i) \tag{4}$$

where S is the set of events (flooded or not flooded), n is the number of events S, and $p_i$ is the probability of any factor in S estimated using $\frac{|S_i|}{|S|}$. The expected information (Entropy) is needed to classify a factor in S. Suppose a factor X consists of m classes $\{X_1, X_2, \ldots, X_m\}$, with each class consisting of flood events and/or not flooding; then, the information on factor X is formulated in Equation (5):

$$\text{Entropy}_X(S) = \sum_{i=1}^{m} \frac{|x_i|}{|S|} \text{Entropy}(S_{Xi}) \tag{5}$$

After that, the information gained on factor X is calculated using Equation (6):

$$\text{Gain}(X) = \text{Entropy } (S) - \text{Entropy}_X(S) \tag{6}$$

By involving the gain information, the gain ratio on factor X can be calculated by finding the ratio between the information gain factor X and the amount of entropy for each class on factor X using Equation (7):

$$\text{Gain Ratio } (X) = \frac{\text{Gain}(X)}{\sum_{i=1}^{m} \text{Entropy}(S_{Xi})} \tag{7}$$

### 2.2.3. Flood Susceptibility Calculation Approach

This flood susceptibility calculation approach used a bivariate statistical approach (FR and WofE) and a random forest (RF) approach as a machine learning application. All the techniques discussed here illustrate the relationship between the flood events and flood conditioning factors.

Frequency Ratio Model

The FR method is the probability of flood events and each factor contributing to flooding events in the study area [17]. The estimated FR value is calculated based on the spatial relationship between the location of the flood incident and each of the factors that caused the flood, which is expressed in Equation (8) [29,30]:

$$FR = \frac{\frac{N_{pix}(FX_i)}{\sum_{i=1}^{m} N_{pix}(FX_i)}}{\frac{N_{pix}(X_j)}{\sum_{j=1}^{m} N_{pix}(X_j)}} \tag{8}$$

where $N_{pix}(FX_i)$ is the number of pixels with flood events in class i, $N_{pix}(FX_j)$ is the number of pixels in factor $X_j$, m is the number of classes in factor Xi, and n is the number of factors in the study area. The flood susceptibility index (FSI) is calculated by adding up all the FR values.

Weight of Evidence

The WofE method is a log-linear bivariate statistical method based on Bayesian theory. This model has been widely developed in spatial analysis for mapping the potential for landslides and floods [16,31]. The mathematical formula given by Bonham-Carter (1991, 1994) was based on the determination of positive weights ($W^+$) and negative weights ($W^-$), as shown in Equations (9) and (10):

$$W_i^+ = \ln \frac{P\{B|A\}}{P\{B|\overline{A}\}} \tag{9}$$

$$W_i^- = \ln \frac{P\{\overline{B}|A\}}{P\{\overline{B}|\overline{A}\}} \tag{10}$$

where $W^-$ is a negative correlation with the weights indicating the absence of effective factors that condition flooding, and vice versa for $W^+$ [16]. P is the probability; ln is the natural logarithm. A and B are the entire area and the incidence of flooding in each factor class, respectively. A and B, respectively, are all events that are not flooded, and all are not flooded in the class of each factor. The weight contrast is the difference between positive and negative weights. The positive contrast value indicates a positive spatial relationship, while the negative one indicates a negative spatial relationship. The magnitude of this contrast value reflects the overall spatial relationship between each class of factors causing flooding. Furthermore, the standard deviation of the contrast is the combined root of the variance of the weights formulated in Equations (11) and (12):

$$C = W^+ - W^- \tag{11}$$

$$S(C) = \sqrt{S^2(W^+) + S^2(W^-)} \tag{12}$$

where $S^2(W^+)$ and $S^2(W^-)$ are the variance of the negative and positive weights. The formula for the variance of the weights is expressed in the following Equations (13) and (14):

$$S^2(W^+) = \frac{1}{N(B \cap A)} + \frac{1}{B \cap \overline{A}} \tag{13}$$

$$S^2(W^-) = \frac{1}{(\overline{B} \cap A)} + \frac{1}{\overline{B} \cap \overline{A}} \tag{14}$$

N is the number of the cell unit. $W_{final}$ is the final weight for the WofE model, which is the ratio between contras and standard deviation:

$$W_{final} = \left( \frac{C}{S(C)} \right) \tag{15}$$

The flood susceptibility index (FSI) is calculated by adding up all the values of $W_{final}$.

Random Forest (RF)

An RF is one of the regression and classification methods introduced by Breiman [32] as a development of the CART method, which was used to improve the classification accuracy. An RF is a classification method that is suitable for large data and has good results [33]. The RF classification method combines independent CART classification trees through a randomization process to form a tree on the sample and factor data (Figure 4). Therefore, this process will produce different classification trees. From a set of decision trees, it is expected to obtain a small correlation between trees to reduce prediction errors [32]. The weight used in an RF is the average impurity decrease.

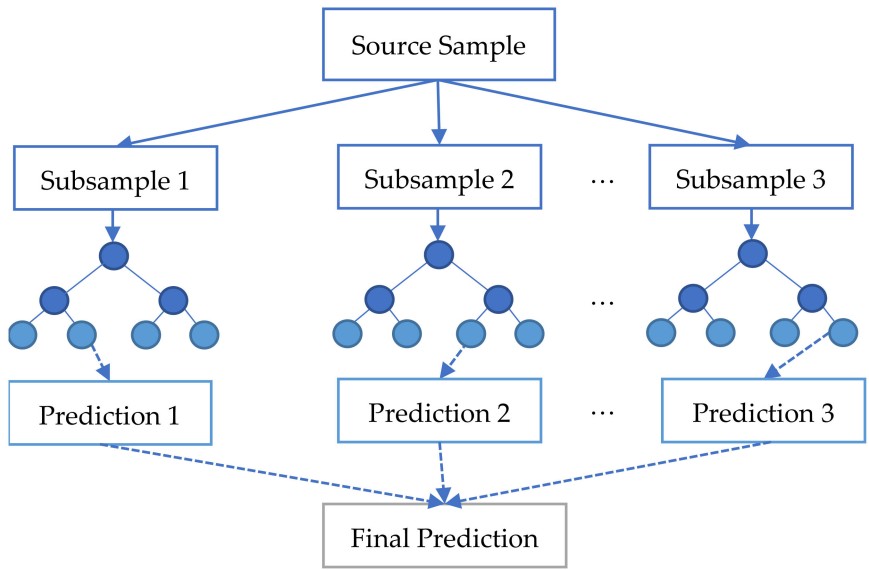

**Figure 4.** Random forest structure [34].

Multi-Layer Perceptron (MLP)

An MLP is a practical approach that describes a complex non-linear relationship between predictors and certain phenomena [35]. The MLP model was developed in the 1960s following an artificial neural network. The structure of the model consists of three layers: the input layer, hidden layer, and output layer. The input is given directly to the output unit via a weight connection. In modeling flood vulnerability with an MLP, the input layer consists of neurons that receive input from flood conditioning factors. The hidden layer consists of neurons that receive input from the input layer and then bring it to the output layer [36]. The output layer is an interaction between binary groups, namely, unflooded and flooded. The neural net training process in MLP uses two main steps [9]: (1) perform a feed-forward network on the input data (flood conditioning factor) through the hidden layer to get the output, and then compare it with the actual value, and (2) adjust the connection weights to get the best results to achieve minimal error. The activation function in the hidden layer neurons is needed to apply non-linear forms to the neural network. It plays a vital role in determining the output of neurons at various values.

$$y_k = \frac{1}{1 + e^{-\Sigma v_{ij} h_j}} \tag{16}$$

$$h_j = \frac{1}{1 + e^{-\Sigma w_{ij}x_i}} \tag{17}$$

$$\sum w_{ij}X_i = w_0 + w_{ij}X_1 + \cdots + w_{Tj}X_T \tag{18}$$

where $W_{ij}$ is the weighting between matrix X and matrix H (the hidden intermediate matrix), $V_{ij}$ is the weighting between matrix H and matrix Y, and $X_T$ is the flooding at location T.

Model Performance Evaluation

The performance of a method can be seen based on the accuracy calculation. Accuracy is the proportion of predictions of an event that correspond to the actual events, which can be described through a summary of the prediction results in the classification processor, commonly called the confusion matrix. The confusion matrix shows the suitability of predictions with actual data.

The accuracy of the flood susceptibility map is calculated based on the difference between the model results and the observations of flood events from the inventory. The accuracy of this flood model is generally described using the receiver operating characteristic (ROC) curve, which is measured from the area under curve (AUC) value [13]. This AUC value is the comparison value between the actual reality presentation and the false presentation of the graph. The ROC curve represents the balance between negative and positive error rates for each possible deal. The process of assessing the accuracy of this model will measure the prediction and success rate as an essential outcome of each program. The AUC value ranges from 0 to 1; if the value is more than 0.5, there is no discrimination. The accuracy level based on the AUC value equal to 0.7 up to 0.8 is in the acceptable category; equal to 0.8 up to 0.9 is included in the excellence category, and equal to or above 0.9 is outstanding [37].

### 3. Result and Discussion

#### 3.1. Multi-Collinearity Test

A multi-collinearity test is used to determine the close relationship between the flood conditioning factors. Using the Pearson correlation calculation, the closeness between factors can be seen with a value between $-1$ and 1 with criteria. If the absolute correlation value is more than 0.8, then it can be said that there is a strong correlation between the two factors [38]. The multi-collinearity test in Table 2 shows that no factors displayed multi-collinearity. Thus, all factors met the requirements for further analysis.

**Table 2.** Multi-collinearity between factors.

|          | Elevation | SPI    | TWI    | Density | Landuse | FA     | Distance | NDVI   | Geology | Soil   |
|----------|-----------|--------|--------|---------|---------|--------|----------|--------|---------|--------|
| Elevation |          |        |        |         |         |        |          |        |         |        |
| SPI      | 0.051     |        |        |         |         |        |          |        |         |        |
| TWI      | −0.185    | 0.376  |        |         |         |        |          |        |         |        |
| Density  | −0.014    | −0.039 | −0.141 |         |         |        |          |        |         |        |
| Landuse  | 0.004     | 0.047  | 0.150  | −0.274  |         |        |          |        |         |        |
| FA       | −0.040    | 0.714  | 0.634  | −0.052  | 0.140   |        |          |        |         |        |
| Distance | 0.087     | −0.043 | −0.012 | −0.118  | 0.119   | −0.074 |          |        |         |        |
| NDVI     | −0.174    | −0.036 | −0.015 | 0.046   | 0.044   | 0.000  | −0.183   |        |         |        |
| Geology  | 0.302     | −0.020 | −0.114 | 0.097   | −0.135  | −0.042 | −0.006   | −0.027 |         |        |
| Soil     | 0.461     | 0.063  | −0.063 | 0.483   | 0.037   | 0.033  | −0.135   | −0.066 | 0.005   |        |
| Rainfall | 0.279     | −0.014 | −0.128 | −0.029  | −0.040  | −0.039 | 0.007    | −0.211 | 0.520   | −0.013 |

#### 3.2. FR and WofE Approach

The flood history data for the study area can be used to predict future flood susceptibility. Therefore, mapping the flood susceptibility in the study area is important to explain the correlation between flooding and the different condition factors. The calculations of probability in the FR and WofE model showed the dissimilarity of the factors that influenced

the flood conditioning, as summarized in Figure 5. Factors that influenced the flooding are indicated by an FR value of more than 1 and the final WofE is positive.

**Figure 5.** Summary of the 11 most important factors of each weighting method.

Based on the FR and WofE weighting results, the two models in Figure 5 have several parameters for flood conditions with different ratings. Regarding the FR, the total value for each parameter in terms of their order of importance for flood conditioning was geology (12.45) with the highest importance, followed by land use (11.87), DR (8.12), soil type (7.93), rainfall 5 years (6.11), NDVI (6.07), RD (6.06), FA (6.04), TWI (5.97), elevation (5.93), and SPI (4.81). Meanwhile, for WofE, the order of the total values for each parameter showed that soil type (30) was the most important, followed by rainfall 5 years (28), elevation (27), land use (25), TWI (15), geology (13), DR (9), RD (8), NDVI (1), FA (0), and SPI (−2).

### 3.3. Information Gain Ratio Test

The results of the selection of 12 factors using the information gain ratio test are shown in Figure 6. The feature selection methods showed that the first most crucial attribute was the elevation and the attribute at the lowest rank was the flow accumulation. Based on the feature selection method, there were seven factors that most effectively influenced the occurrence of coastal flooding, namely, elevation (0.438), geology (0.392), soil type (0.320), land use (0.239), rainfall (0.167), river density (0.130), and TWI (0.123). Meanwhile, the other four factors that were less effective were NDVI (0.028), DR (0.018), SPI (0.014), and FA (0.006).

### 3.4. MLP and RF Approaches

Based on Figure 6, each layer of the gain ratio information results was a flood susceptibility input data model for the MLP and RF approaches. The MLP model used a 0.01 learning rate with 8 hidden layers and optimizes the training data with 10-fold cross-validation. The training process results were the weight for each node in the input layer, hidden layer, and output layer.

In this study, the RF model optimized the training data with a test mode of 10-fold cross-validation and 1000 iterations. An important role in flood conditioning is based on the calculation of the average impurity decrease. Based on the calculation results, each factor had a mean impurity decrease between 0 to 0.24, with the most important factor being rainfall in the class of 70.1–122 mm/h. The lowest rank had a value that occurred in some elevation factors. Several factors lay in the same ranking, as shown in Figure 7.

Factors that influenced flooding were not always the same for each method. In this RF method, the strongest weights were ordered as follows: elevation (0.16), geology (0.14), FA (0.14), RD (0.13), soil type (0.13), DR (0.13), TWI (0.11), land use (0.1), rainfall (0.1), and SPI (0.08).

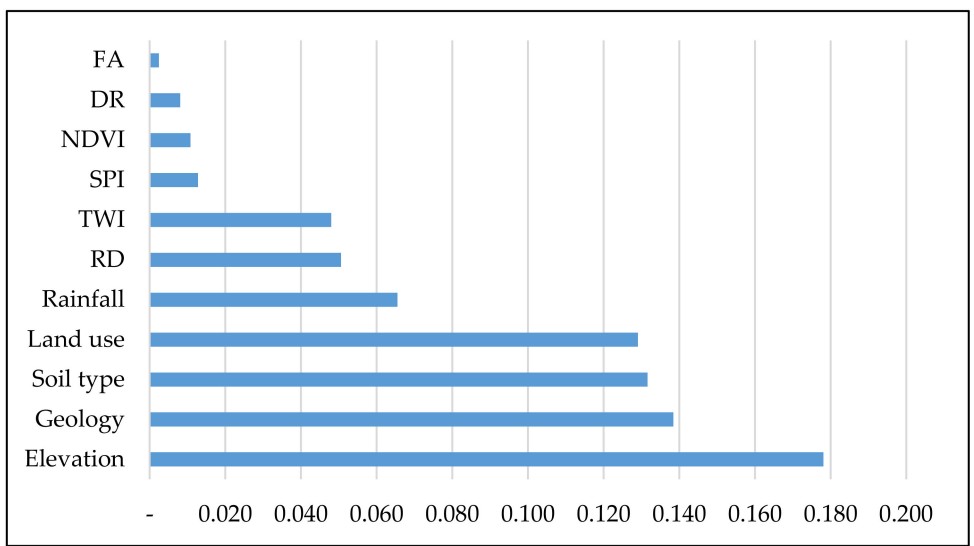

**Figure 6.** The rank of attribute information gain ratio.

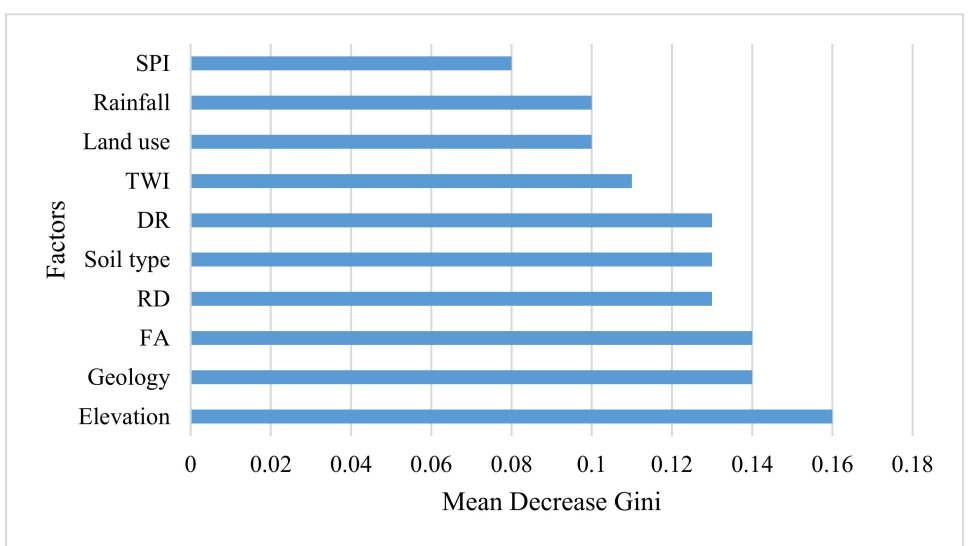

**Figure 7.** Weights of the coastal flood susceptibility determining factors found using RF.

*3.5. Coastal Flood Susceptibility Mapping*

Based on the results of the multi-collinearity test, the coastal flood susceptibility model was implemented in this bivariate statistical approach. Estimates of the coastal flood susceptibility from the four models (Figure 8) were classified into five criteria using the natural break scheme [12,39], namely, very low, low, moderate, high, and very high classes, using reclassification tools on ArcGIS [40]. These five criteria can show a more rational susceptibility level rather than using only three or below. Based on the flood susceptibility map, the results for the high to very-high levels representing coastal areas prone to flooding are marked in blue. Predictions of flood susceptibility models based on FR, WofE, RF, and MLP have identical spatial distributions from these criteria. The percentages of the flood susceptibility index for high to very-high levels (Figure 8) for the FR, WofE, RF, and MLP methods were 19.52%, 20.16%, 21.61%, and 28.01% of the study area, respectively. The

MLP model was more sensitive to capturing high to very high flood susceptibility than the other three.

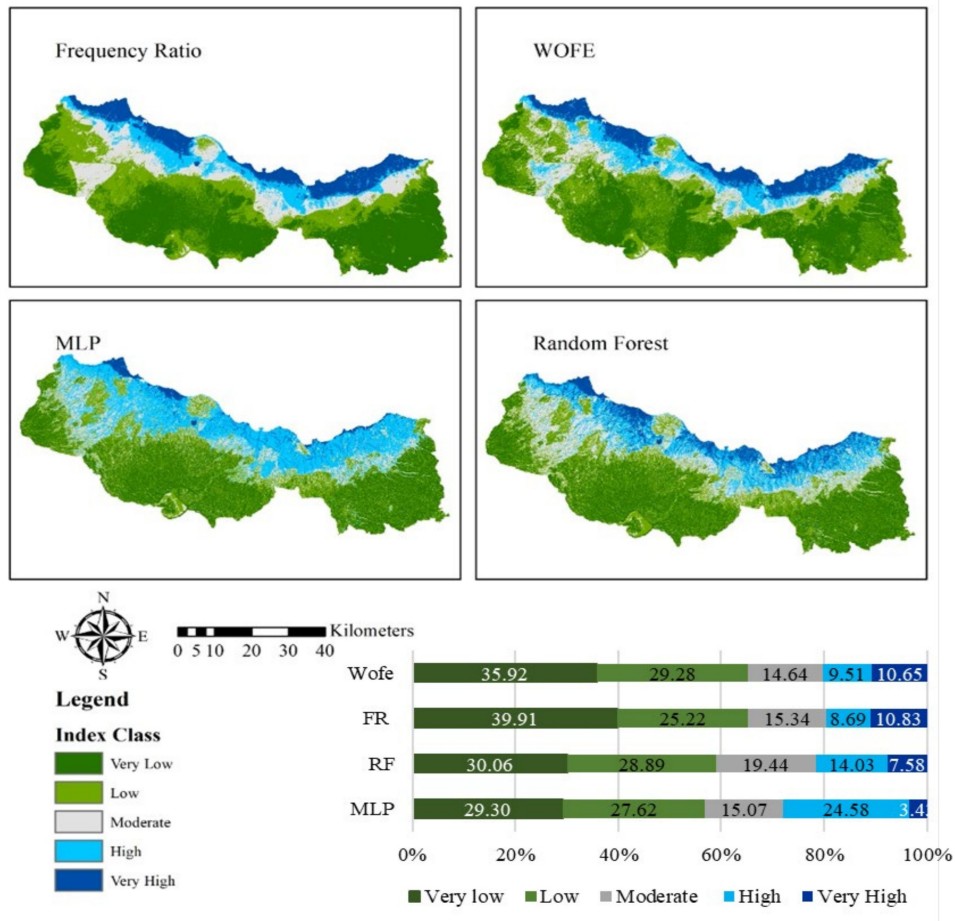

**Figure 8.** Flood susceptibility map obtained using FR, WofE, RF, and MLP methods.

Based on Figure 8, it can be seen that the very high position in the coastal area had an insignificant difference. The WofE and FR models showed the areas with very high levels of flooding that were almost evenly distributed along the coast from east to west. Meanwhile, in the MLP model, the area with a very high level of flooding was only on the west coast and with a high position that extended to the upper part of the southern part of the river.

### 3.5.1. Flood Susceptibility Model Performance

Based on the value of the AUC criteria, the four models showed outstanding performance because they had AUC values > 0.9. This performance model used statistical bivariate model benchmarks (FR and WofE) to assess the machine learning reliability (RF and MLP). The results of the summary of the performance of the training and testing models for each model (Table 3) showed that the best model was the MLP. The lowest AUC value occurred for the WofE model. The AUC values in order from the best training and testing results were MLP (0.967 and 0.956), RF (0.939 and 0.936), FR (0.926 and 0.921), and WofE (0.925 and 0.920). In general, the flood modeling performance with MLP can exceed RF or bivariate statistical models [41]. The modeling of coastal flood susceptibility with machine learning (non-linear) is more reliable than statistical (linear) bivariate models.

**Table 3.** AUC value for flood susceptibility through FR, WofE, and MLP.

| Model | FR | WofE | RF | MLP |
|---|---|---|---|---|
| Training | 0.926 | 0.925 | 0.939 | 0.967 |
| Testing | 0.921 | 0.920 | 0.936 | 0.956 |

The multi-collinearity test of the factors that influence flooding significantly increases the model's predictive capacity [42]. Furthermore, implementing the coastal flood susceptibility model is based on the results of the multi-collinearity test.

3.5.2. Flood Conditioning Factors

The selection of various parameters using IGR, FR, WofE, and the decrease in the mean Gini from RF indicated that the strong correlation in coastal flood conditioning was not the same. However, in general, the most robust weights were elevation, geology, soil type, land use, rainfall, RD, and TWI.

Altitude was the most significant factor for flood conditioning based on the FR, WofE, and RF models. The lowest elevation was a location that was potentially prone to coastal flooding. This was similar to the results of previous studies [23,29,43]. In general, water accumulates from higher elevations to lower areas, and thus, the accumulated water floods a relatively flat area [44]. As with the conditions at the research site, the lowest elevation class was 0–24 m.a.s.l., which was the factor that most conditioned the occurrence of flooding.

The geology of alluvial soil types is one of the factors that play an essential role in flood conditioning. Geological and alluvial soil types have a high level of porosity [45]. Ideally, this type of soil can absorb water so that it does not have the potential for flooding [46]. On the other hand, inundation occurs because the groundwater level in coastal areas is less than 4 m deep. When rainwater accumulates in coastal areas, the groundwater level increases [47], and thus, the chance of flooding increases because the soil layer can no longer absorb water.

Furthermore, the paddy field, plantation, and settlement land-use types have low infiltration rates when compared with forests [48]. The use of paddy fields when compared with forest land will reduce the retention capacity by 33.3% [49]. The small area of forest land will increase the surface runoff. In addition, land planted with a lot of vegetation can slow down the runoff rate so that the possibility of flooding is smaller [50]. Based on FR and WofE, the land-use factors that affected flood conditioning were paddy fields and ponds. Efforts to reduce the risk of flooding can be done by increasing the ratio of forest land area to replace rice fields, especially in areas with steep slopes to reduce the speed of rain runoff that triggers flooding. Optimizing the function of ponds as flood parking is an attempt to reduce overflow.

River density significantly influences the level and intensity of flooding because the river network and the area around the river are very susceptible to manifesting flood events [51,52]. River density, which was a factor of flood conditioning in this study, occurred in the class between 0–2.43; based on the WofE and FR methods, the values ranged from 0–1.49, while for the RF method, the river density ranged from 1.5–2.43. Flood mitigation efforts that need to be done for the characteristics of low river density in coastal areas are anticipating the river valleys that are susceptible to direct flooding. Meanwhile, for coastal areas with high river density values, arranging a drainage network system is necessary to reduce the risk of flooding.

Furthermore, the highest TWI is a factor that affects flood conditioning. A high TWI indicates areas that are prone to saturated soil surface and areas that have the potential to generate runoff [53]. This TWI value has a very high correlation with the flood conditioning factor [17]. Based on Equation (1), the wider the watershed with a smaller slope angle, the greater the TWI value. The wider watersheds in the downstream section contribute to more significant flooding than the narrow watersheds. Therefore, efforts to conserve

catchment areas minimize runoff in large watersheds and maintain corridors by controlling their depth.

The most important factor influencing flooding based on the WofE and FR methods showed a linear relationship with the location of the flood incident (located on the coast). Meanwhile, the factors that affected flooding based on the RF and MLP methods were non-linear relationships with the location of the flood incident. Therefore, the use of RF and MLP methods can describe the geomorphological characteristics of the watershed as the cause of coastal flooding.

These results are consistent with the findings of previous studies [54], which is the slope or altitude influencing the coastal flood. The most common floods were predicted to occur on the flats/in lower areas. Furthermore, based on [55], coastal flood susceptibility mapping using Shannon's entropy model in Oman showed that land use was the most influential factor regarding flooding in the area, which supports the result of this research. Land-use factors have close relationships with flood occurrence in the given area, as they play a major role in water infiltration and surface runoff. Another study [56] that used ensemble machine learning models in flood susceptibility mapping showed that the most vital features of flood modeling were elevation, soil, TWI, lithology, and rainfall.

The limitation of this study was that the bivariate analysis did not have to show a causal relationship. Bivariate statistical models are limited in their capacity to recognize the importance of independent variables because they use a class-based procedure during modeling; thus, to overcome these problems, scientists have developed ensemble statistical approaches [57]. Therefore, in future studies, researchers can use an ensemble method between bivariate analysis with a machine learning method for a comparison to perform the best model. Even though not all cases found that an ensemble is better, it is worth trying. To improve the accuracy, a previous study [58] introduced an ensemble machine learning model that combined MLP, K-nearest neighbor, and RF predictions. Three widely used ensemble techniques are bagging, boosting, and stacking [59].

## 4. Conclusions

Floods in the small watershed of the Pasuruan and Probolinggo regions regularly inundate the north coast and damage homes, rice fields, and highways. The area is very flat, making it difficult to map the floodplain. Four models were tested and the results showed that they yielded excellent AUC values and could represent coastal flood susceptibility modeling. To achieve good model performance, the flood adjustment factors needed to be examined to determine the relationships between them. The findings of this study concluded that machine learning with MLP and RF models was more sensitive to flood vulnerabilities than FR and WofE models, with seven out of eleven factors being very influential, namely, elevation, geology, soil type, land use, rainfall, RD, and TWI. However, this method provided a more comprehensive view of the strong influencing factors of floods toward coastal flood mitigation by comparing the advantages of each model. Flood susceptibility modeling using these four models can fully describe the factors affecting flood susceptibility as a basis for flood mitigation. In addition, multi-collinearity test results for these factors can enhance the model performance.

Machine learning techniques such as RF and MLP are more likely to help provide values for potential flood locations in coastal areas. It is beneficial to consider reducing coastal flooding through spatial planning and predictive actions before flooding occurs in all watersheds. A challenge for further research to increase the accuracy value is to try to classify in other ways to avoid information loss or predict the weights using other methods, such as gradient-boosted trees for ensemble models.

**Author Contributions:** Conceptualization and methodology, E.H. and B.P.; investigation and validation, G.H.; resources and data curation, I.; review, B.P. and W.-K.L.; visualization, W.-K.L.; writing—original draft preparation and writing—editing, E.H. All authors have read and agreed to the published version of the manuscript.

**Funding:** This research was funded by Jember University, grant number 2905/UN25.3.1/LT/2021 and the APC was funded by Jember University.

**Data Availability Statement:** The data presented in this study are available on request from the corresponding author. The data are not publicly available due to this data belongs to Jember University as a research funder.

**Acknowledgments:** The authors want to show appreciation toward Jember University for the funding support for this research.

**Conflicts of Interest:** The authors declare no conflict of interest.

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
