# Peer review of "Assessing Coastal Flood Susceptibility in East Java, Indonesia: Comparison of Statistical Bivariate and Machine Learning Techniques"

_water, doi:10.3390/w14233869_

Round 1

Reviewer 1 Report

The manuscript is very interesting. The study area of the manuscript is very important to be investigated. The authors develop four different models and they found that MLP gives better accuracy. My recommendation is that the article needs Minor Revisions before it can be considered for publication.
Abstract: The authors are encouraged to add the contribution of the study to the abstract.

Introduction: The authors can support the contribution of this study and highlight the superiority of the MLP in another field of engineering by improving the literature section. For example, this study (https://doi.org/10.1080/19942060.2022.2126528)proved that MLP is better for solar radiation prediction. Also, the authors can cite other works that show that MLP is considered best among other ML models.

Methods: The authors are encouraged to add Figures of the used ML models.

Results and discussion: The authors are encouraged to present the limitations of the study at the end of this section. 

Author Response

Dear Reviewer

Thank you for considering our manuscript and giving us the opportunity for further revising the manuscript. Please see the attachment below for the details responses. Thank you.

Reviewer 2 Report

Reviewer comments

Thank you for submitting your paper to Journal of Water. I read carefully manuscript number: water-2015539, the manuscript entitled: "Assessing Coastal Flood Susceptibility at East Java, Indonesia: Comparison of Statistical Bivariate and Machine Learning Techniques". This paper aims to explore four different types of models: frequency ratio (FR) and weights-of-evidence (WofE), random forest (RF), and Multi-Layer Perceptron (MLP) for coastal flood susceptibility assessment in Pasuruan and Probolinggo at East Java region. Factors are selected based on multi-collinearity and information gain ratio to build flood susceptibility maps in small watersheds. In my point of view, the result of this kind of research could be interesting and useful for many applications specifically for the spatial-temporal risk detection, and inundation mapping. The English language is moderate. Some sections of paper require major revisions before any further. I attached my reviewer supplementary comments in the below and attached manuscript file.

1- Abstract

1-1- The abstract section need to complete with more information. The abstract should be improved.

1-2-The concrete finding of this research need to be added to the abstract section.

2- Introduction

2-1- The literature review is too general and thus can’t indicate any novelty of the current study.

2-2- It is better that explain more about the novelty of manuscript in introduction section.

2-3- The manuscript has not quite innovative. Please explain about its novelty.

2-4- Research organization not provided.

3- Research Methodology section were provided in poor way. So needs improvements:

4 -"Results and Discussion" were provided in poor way.

4-1-Results of this study need to be compared with previous research works. Authors are emphatically recommended to provide a new section for this purpose.

5- Conclusion section need rewriting.

6- The punctuation marks in the entire text of the article should be corrected.

Author Response

(The authors gave the same response as above.)

Reviewer 3 Report

Major revisions are needed. Specific comments are attached in the attachment.

Author Response

Dear Reviewer

Thank you for considering our manuscript and giving us the opportunity for further revising the manuscript. Please see the attachment below for the details responses. Thank You. 

Round 2

Reviewer 2 Report

I read carefully manuscript number: water-2015539, the manuscript entitled: "Assessing Coastal Flood Susceptibility at East Java, Indonesia: Comparison of Statistical Bivariate and Machine Learning Techniques". In my point of view,the result of this kind of research could be interesting and useful for many applications specifically for the spatial inundation and risk mapping of multi-index analysis. All previous comments were applied. The authors applied all comments point by point and I confirm their revision. The added information is important and useful and led to improving the manuscript. I accept the revised manuscript in this present form. I concur; the final decision is accepted for publication.

Reviewer 3 Report

The present version can be accepted.